# *Sanguisorba minor* Scop.: An Overview of Its Phytochemistry and Biological Effects

**DOI:** 10.3390/plants12112128

**Published:** 2023-05-27

**Authors:** Alexandra Cristina Tocai (Moţoc), Tijana Kokeric, Septimiu Tripon, Lucian Barbu-Tudoran, Ana Barjaktarevic, Snezana Cupara, Simona Ioana Vicas

**Affiliations:** 1Doctoral School of Biomedical Science, University of Oradea, 410087 Oradea, Romania; tocai.alexandra@gmail.com; 2Department of Pharmacy, Faculty of Medical Sciences, University of Kragujevac, 34000 Kragujevac, Serbia; tijanahf@gmail.com (T.K.); snezanacupara@medf.kg.ac.rs (S.C.); 3Electron Microscopy Centre, Faculty of Biology and Geology, Babeș-Bolyai University, 44 Republicii St., 400015 Cluj-Napoca, Romania; septimiu.tripon@itim-cj.ro (S.T.); lucian.barbu@itim-cj.ro (L.B.-T.); 4Integrated Electron Microscopy Laboratory, National Institute for Research and Development of Isotopic and Molecular Technologies, 67-103 Donat St., 400293 Cluj-Napoca, Romania; 5Department of Food Engineering, Faculty of Environmental Protection, University of Oradea, 410048 Oradea, Romania

**Keywords:** *Sanguisorba minor* Scop., health effects, flavonoids, tannins, electron microscopy, antioxidant capacity

## Abstract

Since ancient times, many plants have been cultivated for their nutritional and medicinal properties. The genus *Sanguisorba* has been used for medicinal purposes for more than 2000 years. These species are distributed in temperate, arctic, or alpine areas in the Northern Hemisphere. Elongated, imparipinnate leaves and densely clustered flower heads are characteristics of the genus *Sanguisorba*. While *Sanguisorba officinalis* L. is mainly known for its significant medicinal applications, *Sanguisorba minor* Scop. is beginning to attract greater interest for its chemical composition and biological effects. Our research collected extensive information on *Sanguisorba minor*, including its history, taxonomy, habitat, and distribution, as well as its bioactive components and biological activities. In addition to electron microscopy of plant parts (root, stems, and leaves), which is described for the first time in the literature in the case of *S. minor*, the study also provides information on potential pests or beneficial insects that may be present. Our goal was to provide important information that will serve as a solid foundation for upcoming research on *Sanguisorba minor* Scop.

## 1. Introduction

The genus *Sanguisorba* includes perennials belonging to the Rosaceae family and includes over 142 species and subspecies distributed throughout East Asia and Southern Europe [1,2,3]. *Sanguisorba officinalis* L. is the most widespread and investigated of all *Sanguisorba* species, but *Sanguisorba minor* Scop. recently has been a target of different research approaches due to the multiple beneficial effects on human health.

The term ‘‘*sanguis*’’ means “*blood*” in Latin, while ‘‘*sorbeo*’’ means “*to soak up*”; therefore, its name justifies the historical use of the *Sanguisorba* species—the plants were used to stop bleeding [4,5].

*S. minor* Scop. is found under trivial names: small burnet (the most common), salad burnet, burnet, pimpernelle, and Toper’s plant [1,6]. *Sanguisorba minor* Scop. owes its name to its size (*minor*—small), since it is considerably smaller in comparison to other burnets [6,7]. *S. minor* is edible (raw or cooked), and it is considered a good ingredient in salads, as its other popular name implies (salad burnet) [1,8,9].

Cultures of the past primarily utilized *S. minor* for therapeutic purposes that were not supported by scientific studies. This study consequently focused on *S. minor* information that was supported by scientific evidence and provides a history of *S. minor.*

*Sanguisorba minor* Scop. was named after Johannes Antonius Scopoli (Giovanni Antonio) (1723–1788), who discovered this species [10]. It was used in traditional medicine in the treatment of conjunctivitis, fever, and diarrhea, as a tincture or infusion [6,11,12,13,14]. The roots of *S. minor* have been used in ancient traditional Chinese medicine to stop internal bleeding and bleeding gums [1,15], while leaves were used for wine flavoring because it was believed that it can protect against contagious diseases [16,17]. American soldiers drank *S. minor* tea before battles in the Revolutionary War in order to prevent bleeding from wounds [17,18]. *S. minor* has been used for treating snake bites, especially the venomous snakes *Vipera berus* and *Vipera ammodytes* in South Europe [8,19].

*Sanguisorba minor* Scop. has been used as a food ingredient because young leaves are edible (cucumber-like taste). For this reason, *S. minor* leaves have been often used in mixed salads or as a flavoring agent in drinks [1]. As mentioned in 1633 in Gerard’s “General History of Plants”, the different medical preparations of the *S. minor* roots were used externally to treat wounds, as well as internally for dysentery and for the regulation of menstruation [12,20]. There is a proverb in Central Italy that praises *S. minor*: ‘‘*L’insalata non è bella se non c’è la pimpinella,*’’ which means “*Salad is not good/if* ‘*pimpinella*’ *is not there*”. Given its high polyphenol content, *S. minor* is considered one of the most promising food medicines [21]. In Romania, *S. minor* is believed to augment appetite, and that is why it may be added to salads, cooked dishes, spinach, soups, or borscht [22].

In addition, *S. minor* leaves have been used to enhance the taste of wine [12,20]. *S. minor* has been used in European folk medicine for healing external or internal bleeding and treating open wounds. *S. minor* is also used in woman’s health for the regulation of heavy or irregular menstruation [1,6].

The literature’s accessible data extensively detail *S. officinalis* L., but there are little data that mention *S. minor* [1,15]. Our search indicates that no reviews that are exclusively about *S. minor* have been published. We have gathered and investigated the efficacy of this plant in order to highlight the phytochemical profile of various plant parts, taking into account the plant’s prospective phytochemical and biological characteristics. Additionally, some information is provided regarding *S. minor*’s pests and beneficial insects.

### Research Methodology

Data on the nutritional and phytochemical composition of *S. minor* and its biological activity were selected using PRISMA Flowchart 2020 based on the suggestion of Page et al., 2021 [23]. Stages and selection criteria, followed by the number of studies used in our review, are presented in Figure 1. The current literature about *S. minor* was collected from PubMed, Scopus, Science Direct, Elsevier, and Google Scholar. The Medical Subject Headings keywords included in the search were as follows: “*Sanguisorba*”, “*Sanguisorba minor*”, “nutrients”, “bioactive compounds *Sanguisorba*”, “phytochemicals *Sanguisorba minor*”, “antioxidant capacity/activity *S. minor*”, ”antimicrobial *S. minor*”, and “anticancer *Sanguisorba*”. Information systematized in the tables was obtained from research articles (in vivo or in vitro studies) between 2017 and 2022. Studies published in languages other than English were excluded. A total of 91 studies were selected and included in this review (Figure 1).

## 2. Taxonomy, Habitat, and Distribution

Family Rosaceae is a moderately large family with 85 genera and over 2000 species [2]. *Sanguisorba* belongs to the family Rosaceae, subfamily Rosideae, tribe Sanguisorbae, and genus *Sanguisorba*. This genus is distinguishable from others by having elongated, imparipinnate leaves and small flowers, tetramerous or trimerous, which lack petals. According to Nordborg’s categorization, both *S. officinalis* L. and *S. minor* Scop. have the chromosome number 2n = 28, 56 [1,2,3,24,25,26,27].

*S. minor* can be found in most parts of Europe, northern Africa, Asia, and America [1,28,29,30]. Nordborg provided a detailed account of the habitat of *S. minor,* which can be found from subtropical to temperate altitudes with a moist, cool climate [17]. *S. minor* prefers slightly dry calcareous soil with limestone rock on the surface or well-drained soil. It is well adapted to grow in nutritionally poor soils [17,29,31,32,33,34].

WFO (World Flora Online) mentions six subspecies of *S. minor*, which are shown in Table 1 (World Flora Online Consortium, http://www.worldfloraonline.org/organisation/WFO) (accessed on 20 April 2023) [35].

The names given to this plant species vary depending on the nation. ‘Salad burnet’ in English, ‘Salvastrella minore, Bibinella’ in Italian, ‘Kleiner Wiesenknopf’ in German, ‘Cebarea’ in Romanian, and ‘Petite pimprenelle’ in French are a few examples (http://www.worldfloraonline.org/taxon/wfo-0001015888#preferredNames) (accessed on 20 April 2023) [42].

## 3. Botanical Characterization: Macroscopic and Microscopic Aspects of *S. minor*

*S. minor* is a perennial species with strongly branched rhizomes in the soil. *S. minor* has long and hard roots, with branched rhizomes that measure about 25–40 cm in length (Figure 2) [6,13,43].

The aerial stem is 20–50 cm high when erect. It can be angular or round; towards the top it is branched, and in the lower part it is hairy (Figure 2A). Leaves are imparipinnate-compound; the basal leaves are 5–15 cm long with 11 to 25 leaflets with a shape ranging from rounded to elongated-oval. The stem leaves can have up to 9–15 leaflets that are 0.7–2 cm long. The width can range from 0.5 cm to 1.5 cm for crenate or sharply serrated edges, respectively (Figure 2A,C) [43,44].

The floral formula of the family Rosaceae is K_(3–)5(–10)_, C_(0–)3–5(–10)_, A_(1–)10–many_, Ĝ_1–many_ [45], but the *Sanguisorba* species has K_4_C_o_A_2-many_ Ĝ_1–many_ [46], where K means a calyx of 4 petals, C refers to the corolla and is not present in these species, A indicates the number of stamens and can be 2 or more (particularly in the *S. minor,* where this number is between 10 and 30), and Ĝ indicates the number of carpels, which can be more than 1. The flowers are grouped in capituliform, terminal inflorescences, with a globose-oval shape and a long peduncle. The upper flowers of the head are female, the middle bisexual, and the lower male (Figure 2B) [44,47]. The receptacle has four uncovered longitudinal stripes on roughly reticulated faces in *Sanguisorba minor* (Figure 2B) [48]. A flower consists of four green or reddish-brown sepals, which are deciduous after flowering and possess numerous stamens [49]. Nectarines are missing. The gynoecium is bicarpellary rarely, and it can have 1–3 carpels [6,47]. Fruits are achenes; they are round, smooth, or broad-winged, and closed in the receptacle (Figure 2B) [50].

The leaves of *S. minor* are green all year round, but the flowers bloom only from May to August, and the seeds appear from July to September [1,4,15,44].

There is no difference in germination percentages between seeds propagated from *Sanguisorba* plants grown in the presence or absence of light [51,52].

*S. minor* is one of the few wild species with edible greenery all year round. The flavor of *S. minor* is nutty, with a hint of cucumber [8,15].

**Figure 2 plants-12-02128-f002:**
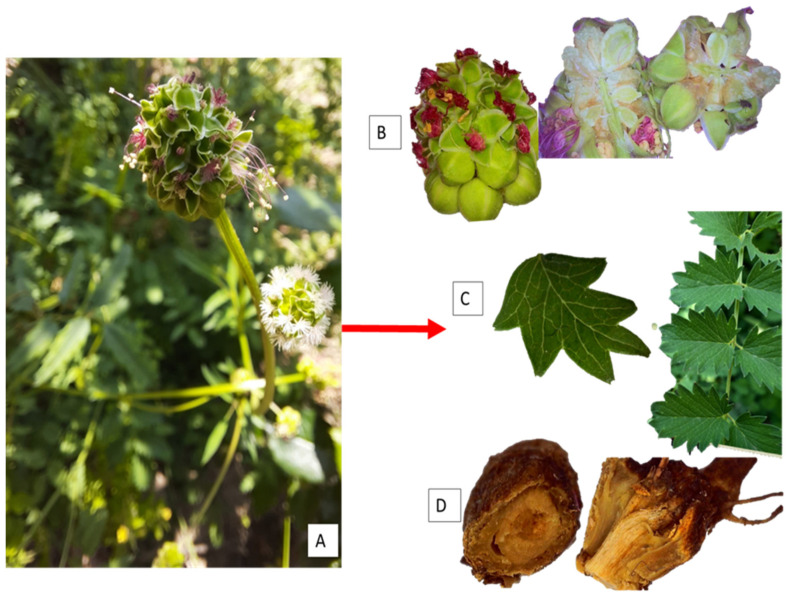
Macroscopic aspects of *S. minor*: (**A**) whole plant, (**B**) flower and a vertical cross-section through the flower (Floral formula: K_4_C_o_A_2-many_ Ĝ_1–many_) [46], (**C**) different types of leaves, (**D**) cross-section through the roots (personal photos).

SEM (Scanning electronic microscope) micrographs of the root, stem, and leaves of *S. minor* are presented in Figure 3A–D. The details related to the samples, materials, and methods for SEM analysis are presented in Appendix A. In the roots, empty areas were seen in the shape of canals, which were probably formed as a result of cells dissolving; in the inner lumen of these canals, raphides were abundant. SEM analysis allowed us to conclude that raphide bundles were frequently associated with the collapsed areas of the phloem (Figure 3A). In the stem micrograph, open stomata can be observed (Figure 3B). The analysis of the leaves (abaxial and adaxial) revealed an epicuticular wax with a crust and granule layers (Figure 3C,D). Based on the wax distribution of the stomata, *S. minor* has a stomata rim and guard cell that are not completely covered by wax and is pore-free on the abaxial and adaxial sides. Because of the distribution on both sides of the stomata, *S. minor* is amphistomatic (Figure 3C,D). Strong environmental correlations exist between the occurrence of amphistomatic leaves (stomata on both surfaces) and hypostomatic leaves (stomata restricted to the bottom or abaxial surface). Amphistomy has the benefit of increasing CO_2_ conductance for photosynthesis, but it may result in ineffective gas exchange if the stomata on both leaf surfaces cannot be independently controlled in response to environmental stimuli [53].

The highlighted microscopic details can provide researchers with some relevant information, as some identifying features were observed by SEM microscopic analysis of the root, stem, and leaves of *S. minor*. These SEM micrographs are special because this is the first time every organ of the *S. minor* plant has been thoroughly detailed. Overall, leaves and surface characteristics are useful for differentiating infrageneric taxa from different subgenera of a genus based on epicuticular ornamentation and stomata. The imaging effect, however, may differ depending on the species of a plant because of variations in leaf structure and composition [54].

## 4. The Nutritional Composition of *S. minor*

The fat, protein, ash, and carbohydrate content of the aerial parts and roots of *S. minor* Scop. were investigated, depending on the type of substrates (peat, peat:perlite (1:1), and peat:perlite (2:1)) by Karkanis et al. (2019) [28]. Perlite is volcanic glass processed through rapid heating and expansion, changing from rock to a foamy texture, while peat is organic matter: a decomposing sphagnum moss. The fatty acid content of *S. minor* roots and aerial plant parts shows a significant difference in their composition. Thus, the portions of aerial plants were rich in linoleic acid (12.9–13.1%), palmitic acid (14.6–15.6%), and linolenic acid (49.4–52.4%), while stearic, tricosylic, lauric, and eicosatrienoic acids were found in small amounts. By contrast, linoleic and tricosylic acids were more abundant in roots (20.5–24.1%), followed by linolenic (12.8%–15.4%) and palmitic acids (11.9–13.1%), while stearic, oleic, dihomo--linolenic, and behenic acids were found in smaller amounts. The linoleic acid and linolenic acid levels were significantly higher in the peat: perlite (1:1) treatment in both aerial parts and roots. Where *S. minor* was grown on peat substrates, the dry weight of the aerial parts was higher; however, the lowest dry weight was observed when the peat: perlite (1:1) treatment was used. Furthermore, the peat: perlite (1:1) treatment yielded the lowest values of root dry weight. Instead, the peat: perlite (2:1) treatment significantly increased the amount of α-tocopherol, glucose, and total sugars, mainly in the aerial parts. The major organic acids that were identified were oxalic acid and citric acid, which were found in higher levels in the roots of the plants than in the aerial sections for all substrate treatments. According to their results, the peat treatment produced better growing conditions from April to May than the other substrate treatments due to its superior water retention properties and the alleviation of stress caused by high temperatures. Karkanis et al. (2019) used various *S. minor* substrate compositions to recommend potential commercial cultivation [28].

In another study, Finimundy et al., 2020 used cultivated *S. minor* under different fertilization rates (using only peat and inorganic fertilizer at different doses). The amount of total phenolic acids in leaves was not considerably affected by fertilizer application, but the amount of the same class of phenolic compounds in roots was greatly enhanced by the administration of the half-rate regime (330 kg/ha). The higher concentration of flavonoids, primarily catechin and its isomers, were found in the roots of plants fertilized at full rate (660 kg/ha), whereas the highest concentration of total phenolic compounds was found in plants fertilized at half rate (330 kg/ha) [55].

The oil content of *S. minor* seed genotypes was between 8.85% and 15.66%. Palmitic acid, oleic acid, linoleic acid, and linolenic acid are the main components of *S. minor* seed oil. The range of palmitic acid, a saturated fatty acid, was 4.55–10.40%. Stearic acid was detected in only three genotypes ranging from 2.28% to 7.90%. Oleic acid was found in all genotypes investigated between 19.57% and 34.34% [50].

The protein content of the leaves and roots of *S. minor* varied between 10.18 and 12.00 g/100 g dw depending on the substrate used. Fructose and glucose were the major sugars identified in both aerial parts and roots, their level being significantly affected by substrate composition [28].

Additionally, *S. minor* has a carminative effect due to its beta-sitosterol, caffeic acid, kaempferol, and quercetin content. *S. minor* is believed to be a tonic plant and contains high levels of beta-carotene and vitamins C and E, as well as having antioxidant qualities that are even greater than those of lettuce or tomatoes due to the presence of polyphenols [21].

In accordance with a study by Ceccanti et al., 2019, *S. minor* leaves from wild edible species are abundant in bioactive components, which remain at initial levels even after being stored. According to the study, this type of plant could be a good substitute for other green plants that are frequently used in the preparation of salads. It also looks promising for the food and flavoring sectors that will require new food ingredients for dietary supplements [29].

The seeds or leaves of this plant are consumed by animals besides humans, including birds, deer, rabbits, and hares, and are a valuable source of food for them [18,20].

Some researchers suggest that the dry powder of *S. minor* could enhance the quality of vegetable oils, particularly those that are lower in antioxidant content, such as maize oil or sunflower oil [19,50,56,57].

## 5. Phytochemical Composition of Botanical Part of *S. minor*

*S. minor* includes a wide range of compounds, such as flavonoids, tannins, triterpenes, phenols, terpenes, and fatty acids [1,25,28,44,58].

Polyphenols, a class of secondary metabolites, play a significant and diverse role in the plant world. The total phenol content of aerial parts, roots, stems, and leaves from *S. minor* is presented in Table 2.

As shown in Table 2, the total phenol content of *S. minor* varies between studies. The factors that influence the amount of bioactive compounds are either related to external factors (soil, climate, and harvest season) or to the type of extraction solvent applied and the extraction technique of these compounds. The investigations measured total phenols using the Folin–Ciocalteu method.

*S. minor* was collected from Romania in September 2019 (during a period of low rainfall) for the investigation by Tocai et al., 2021, [4], whereas Cirovic et al., 2020 [25] obtained *S. minor* from Serbia on sunny days in the spring. Serbia has a hot, humid continental climate or a humid subtropical climate, but Romania has a more unpredictable temperate continental environment that can alter a plant’s features. In both of the experiments, the *S. minor* plant was dried at ambient temperature in a dark, open area [4,25]. The differences in the amount of root total phenolic compounds are caused by variations in the extraction method and the extraction solvent concentration (96% [25] versus 70% ethanol [4]). While Cirovic et al., 2020 [25] extracted the compounds with a solvent reflux at boiling temperature for four hours, Tocai et al., 2021 [4] performed the extraction of phenolic compounds at room temperature. According to the findings shown in Table 2, regardless of the area where the samples were obtained, the solvent used, or the extraction method, the roots of *S. minor* contain the largest quantity of total phenolic compounds, followed by the leaves and stem. The screening of the individual phenols identified in the roots, aerial parts (stems and leaves), or flowers of *S. minor* is shown in Table 3. Flavonoids dominate among the phytochemicals found in *S. minor*. The two primary flavonoids found in the leaves of *S. minor* were apigenin and baicalein. Apigenin has been found to be beneficial for human health; it lowers plasma levels of low-density lipoproteins and inhibits platelet aggregation. Therefore, including apigenin in the diet may be essential [29]. *S. minor*’s leaves and stems are great sources of flavonoids (quercetin, kaempferol, and rutin), which possess antioxidant effects, preventing the oxidative stress associated with aging, cancer, and cardiovascular illnesses [28,44,55,59].

Numerous species of plants contain tannins. Young leaves and flowers are the plant components that have the highest levels of tannins. Due to the phytochemical composition as well as additional elements such as growth stages or environmental conditions (temperature, light, and nutrition), plants have different biological characteristics [60]. *S. minor* has no known adverse effects or contraindications; however, it shouldn’t be used by pregnant women or while breastfeeding because there are insufficient data on the biological effects and toxicity of the plant. Because *S. minor* contains a lot of tannins, it shouldn’t be used continuously for an extended period of time [19]. It may interfere with drugs including fluoroquinolones [61].

In a study [29], *S. minor* seedlings were grown in a nutrient solution, and at two intervals after sowing (15 and 30 days, respectively), the leaves were cut off at the base and subjected to metabolomics analysis. Cutting changed the secondary metabolite profile of *S. minor* in that it enhanced the amount of flavonoids, especially the subclass of flavones, in the leaves that resulted from the second cutting. Instead, the phenolic content and antioxidant capacity of *S. minor* leaves stored for 15 days as fresh-cut products did not significantly change. *S. minor* is a source of phenolic compounds, particularly flavonoids. The leaves of *S. minor* especially are rich in quercetin-3-glucoside and kaempferol-3-glucoside [62]. The aerial parts of *S. minor* (leaves and stems) have a high flavonoid content, with quercetin-3-glucuronide being predominant [28,49].

From an aqueous, ethanolic, whole-plant extract of *S. minor*, Ayoub et al., 2003, isolated and examined the structure of eleven phenols. In addition to the phenols that are typically found in plants (gallic acid, kaempferol, quercetin, ellagic acid, etc.), one of the compounds belongs to the coumarin class. This coumarin dicarboxylic acid derivative is a substance that is extremely infrequently encountered in nature and was isolated for the first time in *S. minor* [63]. The chemical structure of coumarin-3-carboxylic acid is shown in Figure 4.

The composition of the *S. minor* essential oil has only been the subject of one investigation as of yet. In order to explore the chemical composition of the essential oil of *S. minor* leaves from Iran, Esmaeili et al. 2010 observed 17 components, the majority of which were aliphatic hydrocarbons, followed by sesquiterpenes, an oxygenated monoterpene, and an aliphatic aldehyde. Farnesyl acetate, nonadecane, and docosane were the main components, followed by caryophyllene, nonanal, and linalool [64].

Reher et al. (1991) made an interesting observation, pointing out that from a systematic standpoint, *Sanguisorba minor*’s triterpenoid pattern more closely approaches *Sarcopoterium spinosum* (family Rosaceae) than *Sanguisorba officinalis* [58].

## 6. Antioxidant Capacity

Generally, the biological activities of *S. minor* were not investigated extensively. However, according to the literature, *S. minor* exhibits antioxidant [11,56,65,66,67], anti-ulcerogenic [68,69], antitumor [49], antimicrobial [28,44,55,70], neuroprotective [65,71], and anti-inflammatory activity [25,72,73]. Antioxidant activity is the most thoroughly investigated. Papers describing the biological activities of *S. minor* refer to it as a native species growing in Portugal, Spain, Serbia, Romania, and Italy.

The antioxidant capacity of *S. minor* extracts has been confirmed in numerous studies, both in vitro and in vivo. in vitro evaluation was conducted using different antioxidant assays: ABTS (2,2′-azinobis-(3-ethylbenzthiazolin-6-sulfonic acid)) and DPPH (2,2-diphenyl-1-picrylhydrazyl) radical scavenging assays, CUPRAC (cupric reducing antioxidant capacity) assay, ferric-reducing antioxidant power assay (FRAP), total reducing power assay (TRP), and hydroxyl and peroxyl radical scavenging. The summary of the *S. minor* antioxidant capacity results, depending on the sample type and applied method, is presented in Table 4.

The antioxidant capacity of *S. minor* leaves was tested on aqueous and water/ethanol extracts, which demonstrated peroxyl and hydroxyl radical scavenging ability as well as ferric-reducing antioxidant power [56,67]. Ethanolic extract and water decoction of the *S. minor* aerial parts from Portugal in a concentration of 0.1 mg/mL exhibited the ability for DPPH radical neutralization at a high percentage (93% of inhibition). Additionally, essential oil and water decoction of the *S. minor* aerial parts in the same study showed significant inhibition of lipid peroxidation in the carotene-linoleic acid assay (99% and 95%, respectively) [65]. The antioxidant capacity of the methanolic and chloroform extracts of *S. minor* subsp. *muricata* aerial parts was investigated by five different in vitro methods. The results showed stronger antioxidant activity of the methanolic extract than non-polar extracts (chloroform) in all assays (Table 4) [11]. The extraction solvent has a direct effect on the final results of many different antioxidant assays. (Table 4). Generally, methanolic and ethanolic extracts exhibited stronger DPPH and ABTS radical scavenging potential than the chloroform extract. CUPRAC and FRAP assays emphasized a slightly higher antioxidant capacity of ethanolic extract in comparison with methanolic (Table 4) [25].

Multiple in vitro and in vivo studies indicate that total phenols and flavonoids significantly contribute to the antioxidant activity of medicinal plants [74,75]. There are several mechanisms by which plant polyphenols achieve their antioxidant effect. One of the proposed mechanisms is that they act as free-radical scavengers due to their chemical structure and ability for free-radical capture. Another proposed mechanism of action is the chelating of pro-oxidant metals such as iron and copper (Fe^3+^ and Cu^2+^), which prevents them from taking part in free-radical-formation reactions [76,77]. Reducing the activity of phenolic acids and flavonoids, identified in *S. minor* extracts, depends on the number, position, and substitution of hydroxyl groups in the molecule, primarily [28,55]. Many research investigations have shown an association between polyphenol levels and antioxidant activity in medicinal plant extracts [1]. The *S. minor* species’ evident antioxidant capacity has been linked to its high amount of chemical compounds, particularly phenolic compounds [55,56].

The only available in vivo investigation of the antioxidant activity of *S. minor* was conducted on an animal model of sepsis. Considering that oxidative stress is a nonspecific-but-crucial indicator of inflammation and energy disturbances in sepsis, the work of Cirovic T et al., 2020 showed that the ethanolic extract of the *S. minor* subsp. *muricata* root extract affects oxidative stress parameters in rats with induced sepsis [25,74]. Since the focus of up-to-date investigations in sepsis is on antioxidant therapy, as an adjuvant to conventional therapy, medicinal plant extracts with antioxidant and anti-inflammatory activity may play a significant role in the treatment of sepsis [78]. The experiment involved the administration of ethanolic *S. minor* extract to rats with sepsis, orally and intraperitoneally, and monitoring of effects on the level of pro-oxidants (total thiols, TBARS, nitrate and nitrite concentrations NOx, and superoxide anion concentration O^2−^) as well as the activity of superoxide dismutase (SOD). The ethanol extract of the *S. minor* root lowered oxidative stress in rats with sepsis by reducing the plasma levels of TBARS, NOx, and O^2−^, and increasing SOD activity without influencing the level of total thiols [25]. The reason for such a positive effect of *S. minor* on oxidative stress could be phytochemical composition, especially the presence of phenols and flavonoids, which are known to be able to alleviate oxidative stress [28,71,79]. Except for the free-radical scavenger ability, polyphenols can inhibit the activity of certain enzymes responsible for the generation of the reactive oxygen species (xanthine-oxidase and nicotinamide—adenine—dinucleotide—phosphate (NADPH) oxidase). Polyphenols also show an upregulation of endogenous antioxidant enzymes (superoxide dismutase, catalase, and glutathione peroxidase) [76].

## 7. Biological Activities

### 7.1. Antimicrobial Effects

According to the increased need for new antibacterial drugs that can effectively combat drug-resistant infections, plants have been extensively investigated for their antibacterial abilities [80].

Under different growth conditions, Karkanis et al., 2019 examined the antibacterial properties of the root and aerial parts of *S. minor* and found that they depend on the phenolic content. Because roots have higher amounts of phenolic components than aerial extracts, root extracts are more effective antibacterial agents [28].

Another study showed that the bacteriostatic and bactericidal activity of the methanol extract of *S. minor* aerial parts were superior compared to the chloroform extract, with a minimum inhibitory concentration (MIC) range of 0.1–3.13 mg/mL and a minimum bactericidal concentration (MBC) range of 0.39–3.13 mg/mL. Gram-positive bacteria were more successfully combated by *S. minor* extracts (methanol and chloroform) than Gram-negative bacteria. The *Staphylococcus aureus* was the most sensitive (MIC = 0.10 mg/mL, MBC = 0.39 mg/mL) [11].

The study conducted by Cirovic et al., 2020 [25] investigated the antimicrobial activity of ethanol, methanol, and chloroform extracts of *S. minor* subsp. *muricata* roots in the presence of Doxycyclin as a positive control. The chloroform extract of *S. minor* radix showed the strongest antibacterial activity against all examined strains of bacteria: *Bacillus cereus*, *Enterococcus faecalis*, *Staphylococcus aureus, Escherichia coli*, *Pseudomonas aeruginosa*, *Enterobacter aerogenes*, *Proteus mirabilis*, *Klebsiella pneumoniae*, and *Salmonella Enteritidis,* with an MIC range of 0.1–1.56 mg/mL and an MBC range of 0.39–6.25 mg/mL. Generally, *B. cereus* and *S. aureus* were the most sensitive to the chloroform extract (MIC and MBC values were 0.10 mg/mL and 0.39 mg/mL, respectively). In comparison to ethanol extract, the methanolic extract of *S. minor* showed more efficacy against *S. enteritidis*, *E. coli*, *P. aeruginosa*, *B. cereus*, *S. aureus,* and *E. faecalis*.

The antibacterial activity of samples of *S. minor*, both wild and cultivated, was examined against strains of eight different bacteria, including *E. coli*, *K. pneumoniae*, *M. morganii*, *P. mirabilis*, *P. aeruginosa*, *E. faecalis*, *L. monocytogenes*, and MRSA (Methicillin-resistant *Staphylococcus aureus*). All of the plant extracts showed antibacterial activity, with oven-drying samples displaying more antibacterial activity than freeze-drying ones. In all *S. minor* samples, the MIC ranged from 2.5 mg/mL to >20 mg/mL [59].

Other authors validated *S. minor*’s strong antibacterial activity, although they obtained lower MIC and MBC values [28,55]. According to Karkanis et al. (2019), *S. minor* samples (aerial parts and roots) had MIC and MBC values of 0.075–0.45 mg/mL and 0.25–0.60 mg/mL, respectively, against *Bacillus cereus*, *Staphylococcus aureus*, *Listeria monocytogenes*, and *Salmonella typhimurium* [28]. In addition, Finimundy et al., 2020 examined *S. minor* roots and leaves grown under various fertilization regimes and reported MIC and MBC values between 2.31 and 0.44 mg/mL and between 4.61 and 0.88 mg/mL, respectively, against *Staphylococcus aureus*, *Bacillus cereus*, *Mariniluteicoccus flavus*, *Listeria monocytogenes*, *Pseudomonas aeruginosa*, *Salmonella typhimurium*, *Enterobacter cloacae,* and *Escherichia coli* [55].

#### 7.1.1. Antifungal Activity

In the treatment of postharvest fungal infections, *S. minor* may be an effective alternative to synthetic fungicides. In vitro spore germination of *Monilnia laxa*, *Penicillium digitatum*, *Pencillium italicum*, and *Aspergillus niger* was completely inhibited by *S. minor* extracts, while that of *Botrytis cinerea* and *Pencillium expansum* was significantly reduced, according to the studies by Karkanis et al. (2014) and Gatto et al. (2011) [19,62].

Another study showed that the leaves of *S. minor* showed excellent antifungal activity (>80% inhibition) against *Monilinia laxa*, *Monilinia fructicola*, *Penicillium expansum*, and *Penicillium italicum* [81].

Nystatin was used as a positive control for antifungal activity by Cirovic et al., 2020, who reported an MIC of 7.81 mg/mL and an MFC of 15.61 µg/mL. *S. minor* roots showed limited antifungal activity against *Candida albicans* [25].

In another study, Ketoconazole was utilized as a positive control for antifungal efficacy against various fungi with an MIC = 1.25–2.50 mg/mL and MFC = 2.50–5.00 mg/mL depending on the fungus tested by Finimundy et al. in 2020. They found that *S. minor* root and leaf extracts from plants grown under the full fertilization rate regime had similar antifungal activity to *S. minor* root extracts supplemented with fertilizers, with MIC values of 2.50 mg/mL and 5.00 mg/mL, respectively, against *Aspergillus fumigatus* and *Aspergillus niger* [55].

#### 7.1.2. Antiviral Activity

Abad et al., 2000 reported significant inhibition of the herpes simplex virus type 1 (DNA virus) and vesicular stomatitis virus (RNA virus) by *S. minor* subsp. *magnolii* aqueous extract in a concentration range of 50–125 µg/mL. The antiviral activity was evaluated on HeLa (human epithelial cervical carcinoma) cells infected with the viruses in vitro [1,82].

### 7.2. Cytotoxic Activity on Cancer Cells

Plant-derived chemicals have the ability to act as inhibitors of various phases of carcinogenesis and associated inflammatory processes [83].

The ethanolic extract of the whole plant of *S. minor* exhibits strong cytotoxic activity against some cancer cell lines, including HepG2 (hepatocellular carcinoma), and appears to be similarly efficient at stopping the migration of cancer cells caused by plasmin [49].

The cytotoxic activity of *S. minor* roots and leaves was examined by Finimundy et al. (2020) in relation to inorganic fertilization doses. The cytotoxic activity of the root extracts was most effective against the HeLa cervical carcinoma cell line, NCI-H460 non-small cell lung cancer cell line, MCF-7 breast carcinoma cell line, and HepG2 cell line. As opposed to the root extracts, the leaf extracts had less cytotoxic activity against the HepG2 and NCI-H460 cell lines. Their findings showed that the high fertilizer dose (660 Kg/ha) increased the phenolic component content and, as a result, the cytotoxic effects against tumor cell lines [55].

Cuccioloni et al., 2012 confirmed that quercetin-3-glucuronide, isolated from the ethanolic extract of *S. minor,* exhibited cytotoxicity activity in vitro by limiting plasmin-induced migration of MCF-7. Therefore, it seems that quercetin-3-glucuronide may serve as a good starting compound in developing a new pharmaceutical agent that may be used in the treatment of pathological states caused by the unregulated activity of plasmin [49].

### 7.3. Neuroprotective Effects

Polyphenol-rich medicinal plants have considerable antioxidant activity and protect neurons from oxidative damage [84].

*S. minor* aerial parts enhanced antioxidant defense in the brain, and the activity of the antioxidant enzymes (Superoxide dismutase-SOD and catalase-CAT) was significantly decreased in the hippocampus and cortex of the rats in the scopolamine group in comparison to those of the controls (*p* < 0.001) [71].

The Morris water maze test (MWM) revealed that after the treatment with the extract of *S. minor*, especially at the dose of 200 mg/kg, the rats’ capacity to recall the platform’s location improved. Additionally, the administration of the extract enhanced the memory indices in the scopolamine-treated rats during the PA (passive avoidance) test. These results revealed that the *S. minor* extract improved the rats learning and spatial memory capacities since the MWM task often indicated rodents’ ability to memorize spatial locations. Rivastigmine, a common anti-amnesia medication being an AChE (acetyl cholinesterase) inhibitor, produced similar effects in the rats that received it as well [71].

The hydroethanolic solution obtained from the aerial parts of *S. minor* protects against oxidative stress, as well as mitigates the concomitant histopathological changes induced by D-galactose in the liver and brain. In addition, *S. minor* aerial parts reversed the increased activities of AST (aspartate aminotransferase) and ALT (alanine aminotransferase), which may be correlated with liver dysfunction in the presence of D-galactose. Therefore, the antioxidant effects of *S. minor* aerial parts against brain and liver damage, at least partially, were attributed to its protection against oxidative stress. Additionally, phytochemicals such as quercetin and ellagic acid, compounds identified in *S. minor,* were found to suppress AChE activity [85].

The preventive effect of the *S. minor* extract (containing 11.06 0.55 mg GAE/ g extract dw) on beta-amyloid-induced toxicity in primary neural cell culture was pointed out by Akbari et al. in 2019.

When different doses of *S. minor* extract (5–100 μg/mL) and beta-amyloid were applied to cerebellar granule neurons, the cytotoxicity caused by beta-amyloid decreased significantly. The concentration of 75 µg/mL was observed to provide the maximum level of *S. minor* protection. At a dosage of 100 µg/mL, a considerable inhibition of AChE activity of over 80% was also observed. These findings are consistent with the theory that the AChE inhibitory action of the *S. minor* extract contributed to its neuroprotective effect against beta-amyloid neurotoxicity. The authors proposed that the neuroprotective effects were conferred due to the presence of phenolic compounds [86].

### 7.4. Antiulcerogenic Activity

*S. minor* subsp. *muricata* in the form of water decoction is used in Turkish traditional medicine in the treatment of abdominal pain, heartburn, and other gastrointestinal symptoms. Gurbuz et al., 2005 confirmed the significant anti-ulcerogenic activity of orally applied aqueous extract of *S. minor* subsp. *muricata* aerial parts on rats with ethanol-induced ulcerogenesis (62.2% of inhibition). Anti-ulcerogenic activity is a consequence of the presence of saponins, tannins, and flavonoids which exhibit gastroprotective effects [68]. Due to their ability to scavenge free radicals, flavonoids from *S. minor* help to have an anti-ulcerogenic effect. This can be explained because the pathogenesis of damaging and ulcerative lesions of the gastrointestinal tract is greatly affected by the production and excessive accumulation of free radicals [68].

### 7.5. Toxicity Studies

For the acute toxicity test, mice were used in just one trial in which oral doses of *S. minor* hydroalcoholic extract of aerial parts (stem and leaves) were administered at 300, 2000, or 3000 mg/kg. To assess subacute toxicity, the oral administration of *S. minor* hydroalcoholic extract at dosages of 100, 200, and 400 mg/kg was conducted for 4 weeks. In the 14 day observation period, neither mortality nor abnormal clinical signs were observed in animals treated with 300 or 2000 mg/kg extract doses. There was an approximate median LD_50_ (lethal dose) of 3000 mg/kg for *S. minor* hydroalcoholic extract. Within 14 days, the animals treated with *S. minor* hydroalcoholic extract at a dose of 3000 mg/kg showed no changes in behavior patterns. A side effect of *S. minor* hydroalcoholic extract at 3000 mg/kg was muscle twinge and lethargy, which disappeared within 48 h. Skin, fur, eyes, and urine volume, however, were normal. For 4 weeks, *S. minor* hydroalcoholic extract did not cause mortality, morbidity, or toxicity signs at the doses used in this subacute toxicity test. Four weeks after treatment, both treatment and control rats were healthy in terms of behavior and skin, fur, and eyes. The results showed that this plant may be used as an herbal medication and was safe and well-tolerated [87].

There is ample evidence that *S. minor* has a wide spectrum of therapeutic effects from both in vitro and in vivo studies. Appendix A is a list of *S. minor*’s pharmacological effects.

## 8. Insects: Pest or Beneficial

Many ecosystems depend on insects, some of which are helpful, while others are pests [88].

*Philaenus spumarius* is a polyphagous xylem-feeding insect, widespread in the Holarctic, whose nymphs produce a protective foam (spit masses) from their liquid excretion. The eggs usually hatch at the beginning of spring, and the five nymphal stages feed on plant shoots covered by a mucilaginous foam that serves as a barrier that allows the diffusion of O_2_ from the surrounding atmosphere [89,90,91]. Humidity and temperature are particularly limited in the earlier nymphal stages. Adults generally live through one breeding season in the spring/summer, and then in late summer/autumn, the females oviposit and the eggs overwinter in vegetation until they hatch the following spring/summer [89].

Nevertheless, biological, ecological, and ethological information is lacking and dispersed since this species has never been seen as a serious danger to agriculture [89,90].

Several species of *S. minor* were affected by *Philaenus spumarius*, as can be seen in Figure 5.

The most important beneficial insects are bees and bumblebees, which require honey from plants for survival. In their natural habitat, bees are essential for maintaining plant diversity. While the insects consume the plants’ nectar and pollen, the plants benefit from this symbiotic relationship through reproduction and genetic diversity [87,91]. Figure 6 shows how the flowers of *S. minor* have been colonized by bees.

Because the flowers of *S. minor* are filled with pollen, it is known to attract bees, beneficial insects, butterflies/moths, and other pollinators [1,88].

## 9. Conclusions

The taxonomy, phytochemistry, and biological activity of *S. minor* were reviewed. Various biological activities and applications of *S. minor*, including its antibacterial, antioxidant, anticancer, and antiviral properties, have been investigated. In-depth biochemical and clinical investigations are needed to understand the mechanisms by which this plant exerts its multiple benefits and how this plant can be used effectively in the food and pharmaceutical industry. In-depth studies (biochemical and clinical) are required for an understanding of the cellular or molecular mechanisms by which *S. minor* exerts its biological effects. Additionally, further studies on the plant’s pharmacodynamic activities must be performed to establish their efficacy. In order to support the safety of *S. minor*’s main bioactive compounds, investigations on their pharmacokinetics should be performed. In an attempt to establish a connection between the various biological activities and the primary bioactive compounds of *S. minor*, this review detailed the nutritional and phytochemical profile of the herbaceous plant, focusing in particular on the polyphenolic components found in all parts (roots, stems, leaves, flowers, and seeds). As a result, the species *S. minor* offers an invaluable resource of bioactive compounds that must be utilized as successfully as possible to produce either food supplements or pharmaceutical formulations with positive health effects. This updated study, in our opinion, will encourage more research into the phytochemistry and biological impacts of *S. minor*. 

## Figures and Tables

**Figure 1 plants-12-02128-f001:**
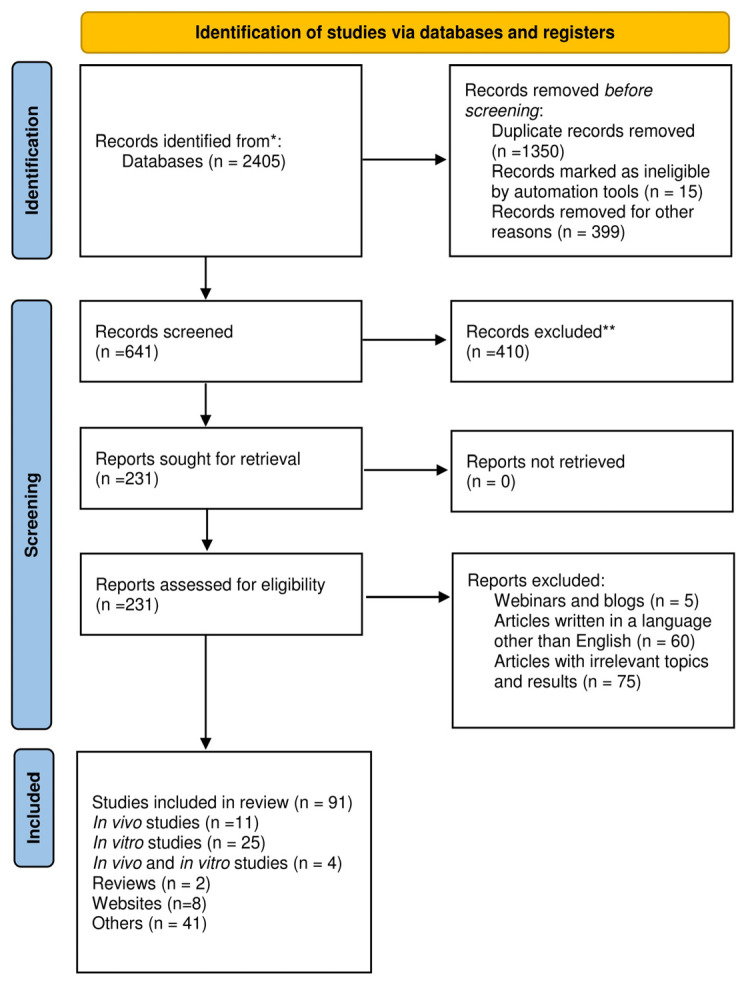
PRISMA 2020 flow diagram of the present review.

**Figure 3 plants-12-02128-f003:**
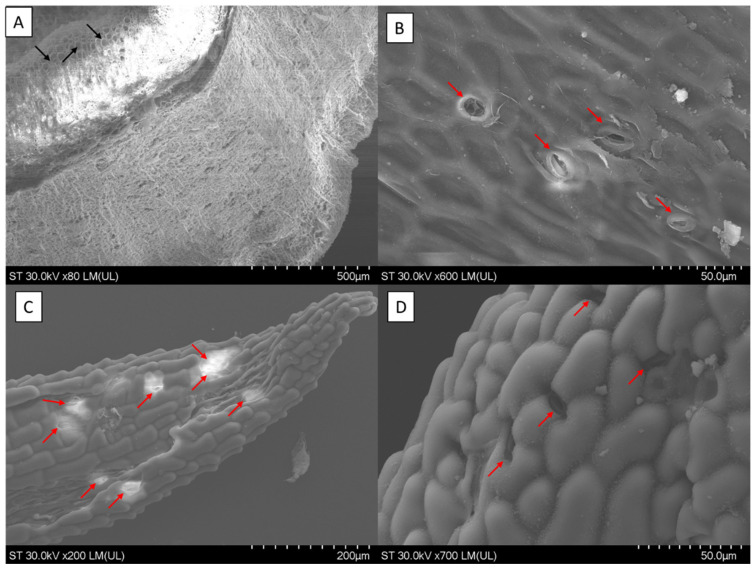
SEM micrographs of *S. minor* root, stems, and leaves: (**A**) *S. minor* root, black arrows denote raphides; (**B**) *S. minor* stem; (**C**) *S. minor* leaf adaxial; and (**D**) *S. minor* leaf abaxial (personal photos). The micrographs (**B**–**D**) provide a detailed view of stomata indicated by the red arrows, while micrograph A shows the empty areas and raphides.

**Figure 4 plants-12-02128-f004:**
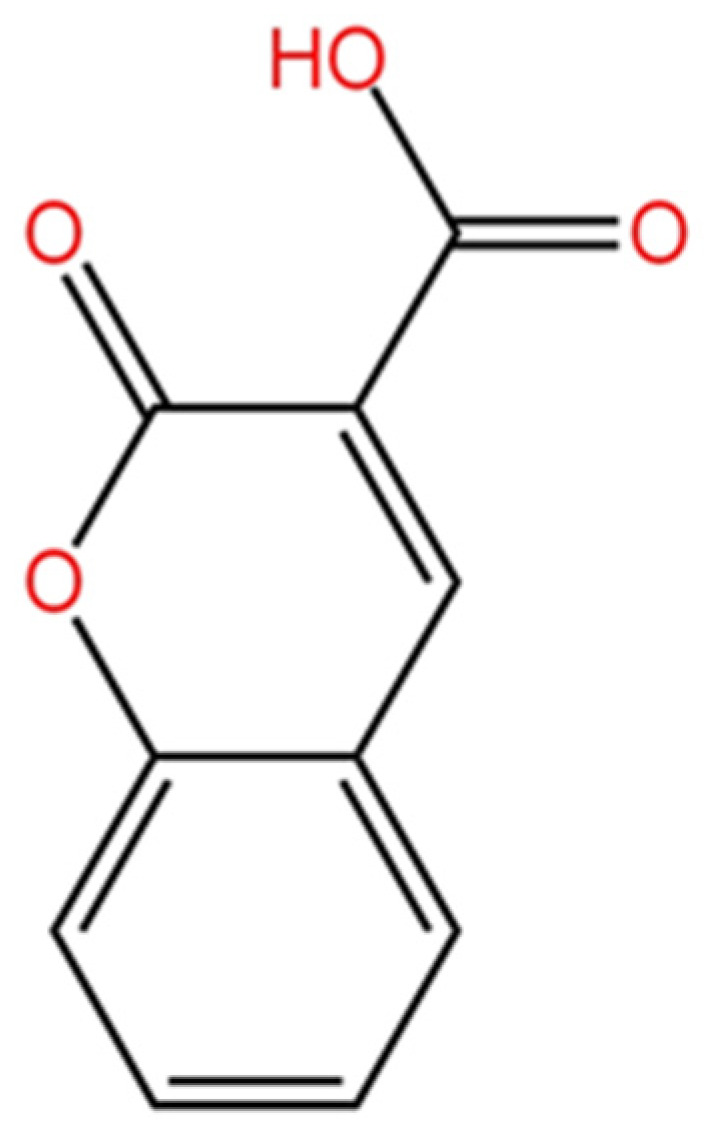
The chemical structure of Coumarin-3-carboxylic acid. (Chemical compound structure was drawn via the ChemDraw tool) (https://chemdrawdirect.perkinelmer.cloud/js/sample/index.html, accessed on 5 May 2023).

**Figure 5 plants-12-02128-f005:**
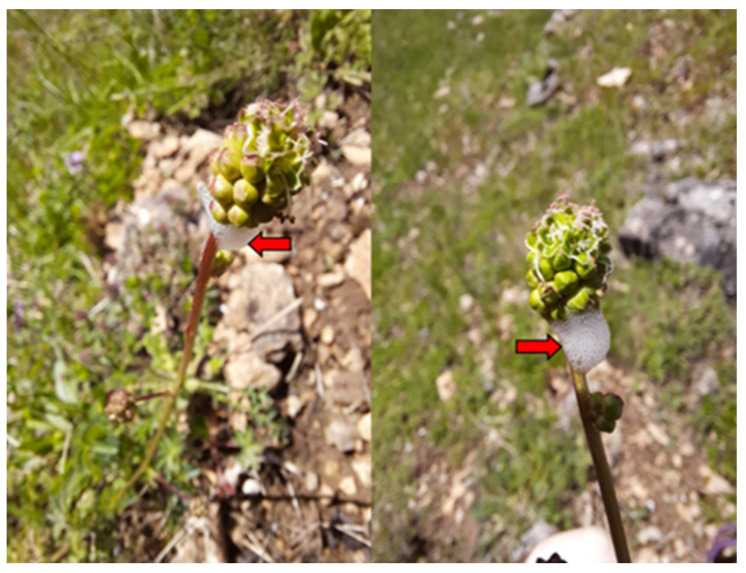
*S. minor* attacked by the *Philaenus spumarius* (personal photos), where red arrows point out spit masses made by this insect.

**Figure 6 plants-12-02128-f006:**
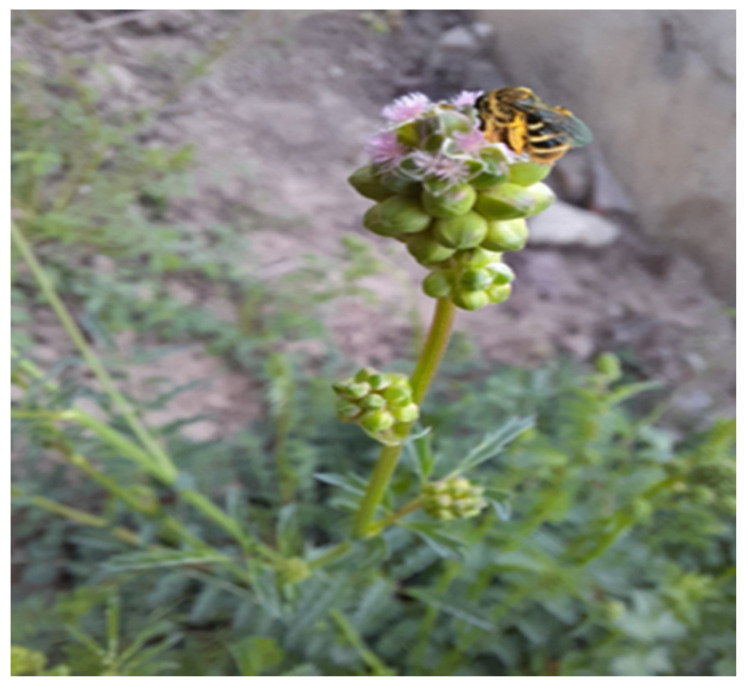
*S. minor* flowers pollinated by *Hymenoptera* species (personal photo).

**Table 1 plants-12-02128-t001:** The subspecies of *Sanguisorba minor* Scop.

1	*Sanguisorba minor* subsp. *balearica*	[36]
2	*Sanguisorba minor* subsp. *lasiocarpa*	[37]
3	*Sanguisorba minor* subsp. *magnolii*	[38]
4	*Sanguisorba minor* subsp. *mauritanica*	[39]
5	*Sanguisorba minor* subsp. *muricata*	[40]
6	*Sanguisorba minor* subsp. *verrucosa*	[41]

**Table 2 plants-12-02128-t002:** The total phenol content of *S. minor* roots, leaves, and aerial parts from the literature of the last few years (2017–2022).

*S. minor* Part	Extraction Solvent	Total Phenols	References
Roots	96% Ethanol	457.45 ± 4.59 μg GAE/mg	[25]
	70% Ethanol	3.89 ± 0.01 mg GAE/g dw	[4]
Aerial parts (stem + leaves)	Methanol	132.80 ± 3.87 μg GAE/mg	[11]
	Chloroform	67.87 ± 0.77 μg GAE/mg	[11]
Stem	70% Ethanol	0.18 ± 0.02 mg GAE/g dw	[4]
Leaves	70% Ethanol	1.19 ± 0.11 mg GAE/g dw	[4]

For GAE-gallic acid equivalent and RE-rutin equivalents, nr—not recorded. Data are expressed as the mean value ± SD. nd—not determined.

**Table 3 plants-12-02128-t003:** The level of phytochemicals (mg/g dw) of tannins, flavonoids, and phenolic acids from *S. minor* roots, leaves, flowers, and aerial parts identified by LC-MS using information from recent literature (2017–2022).

Compounds	Roots	Leaves	Flowers	Aerial Parts (Stems + Leaves)	References
	**Tannins**				
2,3-Hexahydroxydiphenoyl-glucose	15.12 ± 0.21	11.59 ± 0.38	17.82 ± 0.60	nd	[44]
Sanguiin H-10 derivative	5.00 ± 0.06	1.75 ± 0.03	6.76 ± 0.02	nd	[44]
	13.0 ± 0.30	nd	nd	5.14 ± 0.02	[28]
	nd	nd	nd	18.3 ± 0.40	[59]
	6.12 ± 0.05	9.82 ± 0.03	nd	nd	[55]
Punicalagin gallate	28.58 ± 0.04	25.93± 0.02	18.67 ± 0.02	nd	[44]
	21.7 ± 0.70	nd	nd	nd	[28]
	11.5 ± 0.20	nd	nd	nd	[55]
Sanguiin H-1	9.53 ± 0.02	13.46 ± 0.04	26.03 ± 0.05	nd	[44]
Galoyl-bis-hexahydroxydiphenyl –glucoside, isomer 1	5.11 ± 0.22	nd	nd	nd	[44]
	11.1 ± 0.20	nd	nd	nd	[28]
	nd			5.5 ± 0.10	[59]
	12.25 ± 0.02	nd	nd	nd	[55]
Galloyl-bis-hexahydroxydiphenyl-glucoside, isomer 2	4.28 ± 0.03	nd	nd	nd	[44]
	13.13 ± 0.07	nd	nd	nd	[28]
Ellagic acid hexoside	nd	11.49 ± 0.04	0.89 ± 0.03	nd	[44]
	nd	nd	nd	3.8 ± 0.10	[28]
	nd	6.2 ± 0.10	nd	nd	[55]
Ellagic acid pentoside	0.24 ± 0.02	1.16 ± 0.02	1.02 ± 0.05	nd	[44]
	11.66 ± 0.04	nd	nd	nd	[28]
	5.47 ± 0.008		nd	nd	[55]
Pedunculagin	8.0 ± 0.20	nd	nd	nd	[28]
	nd	10.2 ± 020	nd	nd	[55]
Lambertianin C	92.9 ± 0.20	nd	nd	22.3 ± 0.30	[28]
	nd	nd	nd	18.6 ± 0.30	[59]
	nd	9.82 ± 0.03	nd	nd	[55]
	**Flavonoids**				
C-type (epi)catechin trimer	6.27 ± 0.02	nd	nd	nd	[44]
Cyanidin-glucoside	nd	nd	0.13 ± 0.07	nd	[44]
B-type (epi)catechin dimer, isomer 1	3.44 ± 0.03	13.38± 0.03	10.26 ± 0.02	nd	[44]
	37.4 ± 0.90	nd	nd	15.4 ± 0.30	[28]
	8.2 ± 0.80	nd	nd	nd	[55]
Catechin	8.58 ± 0.02	15.42 ± 0.02	6.42 ± 0.01	nd	[44]
	28.0 ± 0.50	nd	nd	20.4 ± 0.10	[28]
Cyanidin-malonylglucoside	nd	nd	0.06 ± 0.02	nd	[44]
B-type (epi)catechin dimer, isomer 2	11.79 ± 0.05	9.77 ± 0.03	3.71 ± 0.03	nd	[44]
	48.8 ± 0.30	nd	nd	16.69 ± 0.07	[28]
Quercetin-galloyl-glucoside	nd	1.95 ± 0.02	1.08 ± 0.01	nd	[44]
Quercetin-glucuronide	nd	20.20 ± 0.02	8.33 ± 0.02	nd	[44]
	nd	nd	nd	18.0 ± 0.10	[59]
	nd	7.6 ± 0.20	nd	nd	[55]
	nd	nd		9.31 ± 0.05	[28]
Quercetin-glucoside	nd	8.17 ± 0.09	8.61 ± 0.04	nd	[44]
Quercetin-galloylhexoside	nd	nd	nd	1.320 ± 0.001	[28]
	nd	0.842 ± 0.04	nd	nd	[55]
Quercetin-O-hexoside gallate (isomer 1)	nd	nd	nd	4.5 ± 0.10	[59]
Quercetin-O-hexoside gallate (isomer 2)	nd	nd	nd	5.3 ± 0.20	[59]
Quercetin-O-pentoside	nd	nd	nd	1.521 ± 0.007	[28]
Kaempferol-glucuronide	nd	6.26 ± 0.02	2.12 ± 0.09	nd	[44]
	nd	0.944 ± 0.02	nd	nd	[55]
Kaempferol-3-O-glucoside	nd	nd	nd	11.3 ± 0.60	[59]
Kaempferol-O-hexoside	nd	nd	nd	9.84 ± 0.04	[59]
Apigenin-O-deoxyhexoside	nd	nd	nd	10.54 ± 0.01	[28]
	**Phenolic acids**				
3-Caffeoylquinic acid (Neochlorogenic acid)	3.42 ± 0.02	3.79 ± 0.04	2.11 ± 0.02	nd	[44]
	nd	nd	nd	22 ± 1.00	[59]
Caffeic acid-glucoside	10.90 ± 0.02	4.73 ± 0.01	2.01 ± 0.02	nd	[44]
5-Caffeoylquinic acid (Chlorogenic acid)	2.10 ± 0.03	3.73 ± 0.01	3.96 ± 0.05	nd	[44]
p-Coumaroylquinic acid	4.08 ± 0.07	8.09 ± 0.07	11.48 ± 0.03	nd	[44]
	nd	6.05 ± 0.25	nd	nd	[55]
Gallic acid glucoside	nd	nd	nd	26 ± 2.00	[59]
Caffeoyl ester (isomer 1)	nd	nd	nd	14.8 ± 0.10	[59]
Digalloyl glucoside	nd	nd	nd	22 ± 1	[59]
	9.5 ± 0.10	7.4 ± 0.30	nd	nd	[55]
Caffeoyl ester (isomer 2)	nd	nd	nd	5.7 ± 0.20	[59]
Ellagic acid	2.85 ± 0.03	nd	nd	nd	[44]
	13.3 ± 0.40	nd	nd	nd	[28]
	4.3 ± 0.10	nd	nd	nd	[55]

Data are expressed as the mean value ± SD; nd—not detected.

**Table 4 plants-12-02128-t004:** In vitro antioxidant activity of *Sanguisorba minor*—literature searching.

Sample	Method	Result	References
Essential oil of *S. minor* aerial parts 0.1 mg/mL (Portugal)	DPPH	11% of inhibition	[65]
	Carotene-linoleic acid assay	99% of inhibition	
Ethanolic extract of *S. minor* aerial parts 0.1 mg/mL (Portugal)	DPPH	93% of inhibition	
Decoction of *S. minor* aerial parts 0.1 mg/mL (Portugal)	DPPH	93% of inhibition	
	Carotene-linoleic acid assay	95% of inhibition	
Aqueous extract of *S. minor* leaves 5 mg/mL (Spain)	Peroxyl radical (H_2_O_2_) scavenging	64.35% of inhibition	[56]
	Hydroxyl radical (OH^•^) scavenging	33.50% of inhibition	
Water/ethanol extract of *S. minor* leaves 0.1–0.01 g/mL (Italy)	Peroxyl radical scavenging	212 LOO•kg of fresh plant	[67]
	FRAP	257 mmol Fe^2+^/kg	
Ethanolic extract of *S. minor* roots (Serbia)	ABTS	77.54 µg TE/mg	[25]
	DPPH	96.51 µg TE/mg	
	CUPRAC	346.49 µg TE/mg	
	FRAP	188.22 µg Fe/mg	
	TRP	1.16 µg AAE/mg	
Methanolic extract of *S. minor* roots (Serbia)	ABTS	76.97 µg TE/mg	
	DPPH	97.29 µg TE/mg	
	CUPRAC	34.35 µg TE/mg	
	FRAP	214.02 µg Fe/mg	
	TRP	1.19 µg AAE/mg	
Chloroform extract of *S. minor* roots (Serbia)	ABTS	46.11 µg TE/mg	
	DPPH	40.51 µg TE/mg	
	CUPRAC	96.80 µg TE/mg	
	FRAP	48.02 µg Fe/mg	
	TRP	0.11 µg AAE/mg	
Methanolic extract of *S. minor* aerial parts (Serbia)	ABTS	77.26 µg TE/mg	[11]
	DPPH	95.06 µg TE/mg	
	CUPRAC	289.09 µg TE/mg	
	FRAP	205.62 µg Fe/mg	
	TRP	0.58 µg AAE/mg	
Chloroform extract of *S. minor* aerial parts (Serbia)	ABTS	53.83 µg TE/mg	
	DPPH	40.31 µg TE/mg	
	CUPRAC	182.90 µg TE/mg	
	FRAP	78.22 µg Fe/mg	
	TRP	0.06 µg AAE/mg	
Ethanolic extract of *S. minor* roots (Romania)	DPPH	92.93% of inhibition	[4]
	FRAP	10.81 μmol TE/g	
Ethanolic extract of *S. minor* stems (Romania)	DPPH	0.32% of inhibition	
	FRAP	0.16 μmol TE/g	
Ethanolic extract of *S. minor* leaves (Romania)	DPPH	43.15% of inhibition	
	FRAP	2.88 μmol TE/g	

Where, DPPH (2,2-diphenyl-1-picrylhydrazyl) radical scavenging assays; ABTS (2,2′-azinobis-(3-ethylbenzthiazolin-6-sulfonic acid)); CUPRAC (cupric reducing antioxidant capacity) assay; FRAP (ferric-reducing antioxidant power) assay; TRP (total reducing power) assay; LOO•, trapping peroxyl radicals; TE, Trolox equivalents; AAE, ascorbic acid equivalents; Fe-Fe (II) equivalents.

## Data Availability

Not applicable.

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
