# Peer review of "Sanguisorba minor Scop.: An Overview of Its Phytochemistry and Biological Effects"

_plants, 2023, doi:10.3390/plants12112128_

Round 1

Reviewer 1 Report

This review on Sanguisorba minor could be interesting to those wishing to know more about this medicinal plant or entering the field of research if it presented the data in a critical way. As it is, it presents a snapshot of the literature mainly on its polyphenolic composition and some biological properties without a critical evaluation and sometimes even with the contradictions presented in the research papers but without commenting on them. Furthermore, it does not identify the areas that should be investigated in the future and the gaps in knowledge about this plant. The SEM micrographs are an asset of this review. 

In detail, I give some recommendations to the authors that might help them.

1. After the correct botanical name is first given (Sanguisorba minor Scop.), all the following references can be as S. minor (not S. minor Scop.)

2. There are many synonyms and subspecies of S. minor presented in https://wfoplantlist.org/plant-list/taxon/wfo-0001015888-2022-12?page=1

The authors could broaden their research using synonyms and should present the subspecies in Section 2

3. line 46. "is presented" instead of "has been"

4. line 57: venomous instead of venimious?

5. line 66: the word potential (=possible) seems inappropriate. 

6. line 75: biological properties instead of medical observations?

7. Figure 1. The diagram does not present information on the databases they used, and on what criteria they excluded studies. What are the other reasons that guided them to exclude studies?

8. 1.Section 5. The authors had better present studies on the essential oil composition since later in Table 3 they refer to it.

8.2. They had better refer to the yields of extraction from the different parts.

8.3. Table 1. Although it is not data provided by the authors, it makes me wonder why the total phenol content is lower than the total flavonoid content in some instances. Both assays are not selective, but in most studies, the total phenol content is higher.

8.4. Moreover, they should present the studies that describe the isolation of natural products and therefore their chemical structures (e.g. Ayoub, N. A. (2003). Phytochemistry63(4), 433-436./Reher, G., Reznicek, G., & Baumann, A. (1991). Planta medica57(05), 506-506.).

8.5. They should also present the level of confidence we should place in the results presented in Table 2, i.e. have the compounds been isolated or is it a tentative identification by LC-MS. How have the quantitative results been obtained? Have the researchers used standards for every compound or are they expressed as equivalents of another compound? Does the abbreviation nd in the table always mean not detected (studied but no response) or it might also mean not determined?

9. In lines 224-225 they state "Concentration of polyphenols is much higher in stems and leaves compared to roots" and then in lines 317-318 "Since roots have higher concentration of phenolic compounds than the aerial part..." Which is true?

10. Section 6.3. The word antitumor and anticancer are wrongly used. Antitumor refers to compounds that stop tumor growth in animals and anticancer to compounds that have proven clinical benefit. What the authors present in this Section are results of experiments in cell lines (in vitro) and thus the word cytotoxic describes them better. The authors had better comment on the cytotoxicity of those extracts since they contain mostly polyphenols and the GI50 concentrations are in mg/mL level. What does the word fertiliser in lines 364 mean?

11. In lines 412-413 they attribute the antiulcerogenic potential to tannins and flavonoids whereas in Table 4 to saponins and tannins. Which is true?

12. line 413. It is well known that tannins inhibit enzymes in a not selective way. Is there something special about this reference?

13. line 424. Why do the authors say that it has not been tested for toxicity whereas in the next lines they present such studies? Those studies are very important and had better not have been presented in others...

14. lines 440-450. There is nothing special about those observations. They might be discussed earlier.

15. Conclusions should be enriched with their view on the current state of knowledge and present future directions as promised in the abstract. For example: what are the differences among subspecies, among different geographical locations, ....?

Reviewer 2 Report

The article gives an overview of the bioactive compounds and biological activities of Sanguisorba minor. The topic is interesting and the overview is quite well organized. However, in my opinion, the review lacks a critical evaluation of the literature, especially the sections related to biological activities.

Reviewer 3 Report

The review entitled “Sanguisorba minor Scop.: an overview of its bioactive compounds, biological and pharmacological effects has been submitted to the special issue  “Bioactive Compounds from Medicinal Plants and Plant-Based Foods: Advances and Opportunities 2.0”

The manuscript falls in the scope of Plants and in the scope of the mentioned SI. It reports the ethnobotany, photochemistry, and biological activities of Sanguisorba minor (Rosaceae), collecting data from research articles between 2017 and 2022, comprising a total of 81 references.

In this referee's opinion, the manuscript is not very well written and some parts need a profound revision. Moreover, the originality of the manuscript is low, since there are two recent reviews (2017 and 2021) describing the Sanguisorba genus and highlighting this species as well.

Therefore, I recommend this paper be reconsidered after a major revision.

The terms “biological”, “pharmacological” and “pharmaceutical” activities (or effects) are used in a random manner throughout the text. However, they are not synonyms. What are the differences between these terms? In this referee's opinion, since the results described rose from biological screening methods on S. minor extracts, the most correct term should be “biological activity”; Please correct this in all text, titles, abstract, and tables.

Title: the title is redundant and should be revised. Suggestion: Sanguisorba minor Scop.: an overview of its phytochemistry and biological effects.

Abstract: line 25 – do not use the trivial name of the plant without previously mentioning it.

Line 31 – Rosaceae family

Lines 44 -45 – the sentence needs revision

Please, use only one trivial name throughout the text (or S. minor)

Line 73 – 74 – references needed. Moreover, since there´s a recent review on the Sanguisorba genus published in Frontiers of Pharmacology (2021; 12: 750165) and also focusing on S. minor phytochemistry and biological activities, what is new about this article? What is the added value of it compared with the one recently published? In summary, the authors must clearly explain why they chose S. minor to do a literature review. What kinds of articles are already published? Is there some review on the subject or is this the first one? What is the main contribution of this review?

Lines 139 – 168 - please clarify – is this a research work from the authors? And do you want to include it in a systematic review? Is it already published elsewhere? This must be justified in the text.

Lines 170 – 189 – How does the substrate composition influence the nutritional components? This part needs a proper discussion, please revise.

 Lines 192 – 193 – “… contribute to its healing properties”. It´s an overstatement, please delete.

Since several compounds, including terpenoids, steroids, lignans (etc) were identified in S. minor (lines 191 -192) what is the reason to only describe tannins, flavonoids, and other phenolics in Table 2?  Like it is it seems that only these types of compounds are important and responsible for the described biological properties.

 Table 1 – Some results must be critically discussed.  For example, how could you discuss the high difference between the results obtained in references 25 and 4, for the same type of extract?

Line 207 – gallic acid is not a flavonoid.

Lines 214 – 218 – this paragraph needs revision.

Line 223 – fluoroquinolones?

Line 230 – please revise the title of the section

The antioxidant results described are all obtained using chemical assays; this subsection is not well included in the biological properties section.

Lines 263 – 266 – redundant paragraphs

Line 269 – emphasized?

Line 283 – is phenolic compounds considered phytonutrients?

Lines 280 – 288 – are very confusing and highly redundant. Please revise.

Line 310 – Antimicrobial effects should include antibacterial, antifungal, and antiviral activities. Therefore, lines 452- 468 (antifungal) and 469 – 477 (antiviral) must be located here. Why do you choose to describe these activities as 6.6 Others?

Lines 311 – 318 – revise the paragraphs

Please confirm the MIC values and units of antibacterial activities.

Lines 337 – 339 and 358 – what type of extracts were studied? What were the solvents used?

Lines 361 – 363 – Hela cells showed activity against the extracts? Makes no sense. Revise all section, please.

Lines 364 – 366 – what do you mean by “both fertilizer treatments”?

Lines 402 – 406 – were these compounds isolated or just identified in the extracts? Moreover, this part of the text is a copy of the discussion of reference 70, which referred to four additional references (20 and 37-39, see reference 70), one of them (20) being a review article. These should be revised and corrected!

Lines 414 – 418- this kind of sentence is somewhat very speculative and should be avoided. Please revise.

Lines 421 – 423 – this sentence makes no sense.

Lines 440 – 451 – locate this section as part of section 4. Avoid repetition.

Antifungal and antiviral activities should be relocated on 6.2 as previously suggested.

Line 454 – which authors?

Lines 467 – 468 – revise concentrations and units. 12.5 mg/mL is too high to be considered a moderate antifungal activity. What is the cutting-point value? What about the activity of the positive control? The results should be critically discussed.

Why is Table 4 located after the presentation of the biological activities? Is it a summary of what is already presented? Why is the title “pharmacological activity”? Revise the entire table; for a better understanding of the text and for clarity reasons you should follow the order of the biological activities described in the main text. Avoid repetition with Table 3.

Lines 486 – 514 – Section 7 refers to very general considerations and doesn´t bring any added value to the paper; therefore, it should be deleted.

Revise the conclusion. There´s no application of S. minor in the pharmaceutical industry; at maximum it can be stated that the described biological activity corroborated the use of the species in folk medicine.

In summary, the paper must be thoroughly revised and improved. It´s mandatory to make a critical assessment of all the collected data. An improved discussion and conclusion will help the reader to make decisions about future studies' directions, and what is already shown to be promising and not promising. Some clear conclusions for future work should also be given.

Round 2

Reviewer 1 Report

The authors have made an extensive revision according the reviewers' comments.  I still believe that they should include the synonyms as search terms, but even now it can be accepted.

1. line 217: rewrite for clarity

2. lines 238-242. The content should be changed since the authors do not present the total flavonoid content and thus the references to it should be deleted.

3. line 297. Why not show the structure? That is  just a suggestion.

4. lines 431-434 are irrelevant in the content of that section ( a paragraph about nutritional values within a section about antifungal activity)

5. lines 533-534 and Table 5. First they seem not properly placed in the Toxicity section, and second, they seem redundant because they have been presented. The authors could keep them in Supplementary.

6. Lines 574-576 make no sense. Try to rephrase and be more accurate.

Reviewer 2 Report

The text has been improved. The paper can now be accepted for publication.

Reviewer 3 Report

The authors addressed the suggestions and revised the paper accordingly.

Minor comments:

Lines 50 and 80 - are these subtitles?

Line 75- "our search"

Lines 110 -113 - This information is already described in Lines 43-44. Please revise

table 2 - only describe the total phenols content. Where are the flavonoid content values? (line 238). Regarding the units of TPC values, are they expressed in GAE?

Lines 248 -250 - please revise the sentence

Line 436 - correct the units of MIC - micrograms/mL (mg/mL)

Title of Table 3 - if the authors quantified the compounds they must have standards, therefore the compounds were not tentatively identified. 

Importantly, an English revision is mandatory.

Round 3

Reviewer 1 Report

The authors have revised the manuscript an addressed all points.

Reviewer 3 Report

The authors made revisions and improved the paper. It can be published in the current form-